# Cylindracin, a Fruiting Body-Specific Protein of *Cyclocybe cylindracea*, Represses the Egg-Laying and Development of *Caenorhabditis elegans* and *Drosophila melanogaster*

**DOI:** 10.3390/toxins17030118

**Published:** 2025-03-01

**Authors:** Yamato Kuratani, Akira Matsumoto, Ayako Shigenaga, Koji Miyahara, Keisuke Ekino, Noriaki Saigusa, Hiroto Ohta, Makoto Iwata, Shoji Ando

**Affiliations:** 1Faculty of Biotechnology and Life Science, Sojo University, 4-22-1 Ikeda, Nishi-ku, Kumamoto 860-0082, Japan; yamato.kuratani@gmail.com (Y.K.); miya0320@bio.sojo-u.ac.jp (K.M.); ekino@bio.sojo-u.ac.jp (K.E.); noriaki@bio.sojo-u.ac.jp (N.S.); hiohta@bio.sojo-u.ac.jp (H.O.); 2Faculty of Medicine, Juntendo University, 1-1 Hiraka Gakuendai, Inzai 270-1606, Japan; amatsumo@juntendo.ac.jp; 3Institute of Health and Sports Science & Medicine, Graduate School of Health and Sports Science, Juntendo University, 1-1 Hiraka Gakuendai, Inzai 270-1695, Japan; ayamatsu@juntendo.ac.jp; 4IMB Co., Ltd., 1070-10 Hitotsugi, Asakura 838-0065, Japan; iwata@mush-imb.co.jp

**Keywords:** antifungal protein, basidiomycetes, *Cyclocybe cylindracea*, *Caenorhabditis elegans*, cylindracin, *Drosophila melanogaster*, host defense activity

## Abstract

Mushrooms are a valuable source of bioactive compounds to develop efficient, secure medicines and environmentally friendly agrochemicals. Cylindracin is a small cysteine-rich protein that is specifically expressed in the immature fruiting body of the edible mushroom *Cyclocybe cylindracea*. Recombinant protein (rCYL), comprising the C-terminal cysteine-rich domain of cylindracin, inhibits the hyphal growth and conidiogenesis of filamentous fungi. Here, we show that rCYL represses the egg-laying and development of *Caenorhabditis elegans* and *Drosophila melanogaster*. The feeding of rCYL at 16 µM reduced the body volume of *C. elegans* larvae to approximately 60% when compared to the control. At the same concentration, rCYL repressed the frequencies of pupation and emergence of *D. melanogaster* to 74% and 40%, respectively, when compared to the control. In virgin adult flies, feeding of rCYL at 47 µM substantially repressed the frequency of egg-laying, and the pupation and emergence of the next generation, especially for females. These inhibitory effects of rCYL gradually disappeared after ceasing the ingestion of rCYL. The use of fluorescence-labeled rCYL revealed that the protein accumulates specifically at the pharynx cuticles of *C. elegans*. In *D. melanogaster*, fluorescence-labeled rCYL was detected primarily in the midguts and to a lesser degree in the hindguts, ovaries, testes, and malpighian tubules. rCYL was stable against trypsin, chymotrypsin, and pepsin, whereas it did not inhibit proteolytic and glycolytic enzymes in vitro. rCYL oligomerized and formed amyloid-like aggregates through the binding to heparin and heparan sulfate in vitro. These results suggest that rCYL has potential as a new biocontrol agent against pests.

## 1. Introduction

Host defense is essential for the life cycle of many organisms. One conserved defense mechanism is the production of toxic substances such as secondary metabolites [1,2] and peptides/proteins [3,4] that are active against microbial pathogens and predatory animals. The exploration of previously uncharacterized natural biotoxins and elucidation of the mechanisms underlying their biological functions are ongoing areas of research worldwide, with the aim of developing safe, effective medicines and environmentally friendly agrochemicals.

Insect pest control is a major issue for human health and food security [5,6]. Some insects are vectors of pathogens that cause serious diseases [5,7]. Insect pests can also significantly threaten agricultural production [6]. Current pest control measures largely depend on the use of conventional chemical insecticides, which can result in a genetic selection pressure that potentially leads to the development of insecticide-resistant insects, as well as concerns over the deleterious effects on human health and the environment. Alternative approaches using natural insecticide toxins as novel biopesticides are therefore attracting increased interest [8]. *Bacillus thuringiensis* toxins are environmentally safe insecticidal proteins that have been used as biopesticides with great commercial success [9,10,11]. Some *B. thuringiensis* toxins have been reported to bind target receptors such as cadherin, ADAM metalloproteases, or GPI-anchored alkaline phosphatases on the brush-border membranes of insect midgut epithelial cells and then form pores (channels) in the phospholipid bilayer to cause cell death via colloid-osmotic lysis [10,11]. Defensins are small (<10 kDa), cationic, cysteine (Cys)-rich proteins produced by essentially all eukaryotes that exert antibiotic activities [12,13]. Some plant defensins, such as BrD1, VuD1, and VrD1, show insecticidal activity through the inhibition of digestive enzymes such as trypsin, chymotrypsin, and α-amylase [14,15,16]. The transgenic production of the recombinant defensins in plants affords resistance against insect pests [14]. Besides defensins, several plant α-amylase inhibitors [17], as well as a trypsin inhibitor, cospin, from the mushroom *Coprinopsis cinerea* [18] show insecticidal activity. Lectins are carbohydrate-binding proteins and show toxicity against a broad spectrum of insects, such as Coleoptera, Diptera, and Lepidoptera. Lectins increase mortality, repress development and/or adult emergence, and reduce fecundity [19]. Some lectins exert their toxic effects via binding to the peritrophic matrix (PM), a non-cellular semipermeable layer that lines the midgut lumen, or binds to the brush-border microvilli of the midgut epithelial cells, disrupting nutrient absorption and assimilation [19,20]. Recently, some mushroom lectins were reported to show entomotoxic and nematotoxic activities [21,22]. Ribotoxins from fungi, such as ageritin, cleave a single phosphodiester bond located within the universally conserved loop of 23–28S rRNAs and consequently inhibit ribosomal protein synthesis, thereby playing a role in the defense against insects [23]. 

Mushrooms are rich in useful bioactive compounds with medicinal and agrochemical potential, such as antifungal, antibacterial, antiviral, insecticidal, and nematocidal properties [24,25,26]. Cylindracin is a small Cys-rich protein that is specifically expressed at the pileus surface of the immature fruiting body of the edible mushroom, *Cyclocybe cylindracea* [27]. In a previous study, we prepared recombinant cylindracin (abbreviated here as rCYL) and observed its antifungal activity against filamentous fungi and its inhibition of hyphal growth and conidiogenesis [27]. rCYL contains an extra *N*-terminal sequence, Gly–Pro, that is followed by the C-terminal Cys-rich domain (59 amino acids) of cylindracin. rCYL harbors eight cysteines in the Cys-rich domain, which form four intramolecular disulfide bridges, and is rich in basic residues. rCYL has a calculated molecular weight of 6398.18 and a pI (isoelectric point) of 8.07. Because cylindracin is specifically expressed in the immature fruiting body that initiates spore formation, it might play an important role in the host defense of the fruiting body against predatory animals and pathogenic microorganisms. The aim of this study was to evaluate the potential of rCYL as a biocontrol protein against predatory insects and nematodes. We examined the toxic effects of rCYL using two animal models, *Caenorhabditis elegans* and *Drosophila melanogaster*, because nematodes and flies represent ecologically relevant phyla of fungal predators [28,29]. Our findings show that rCYL represses the egg-laying and development of *C. elegans* and *D. melanogaster*, indicating the potential of rCYL as a novel pest control agent.

## 2. Results

### 2.1. Inhibition of the Development and Egg-Laying of Nematodes

To test the inhibitory activity of rCYL on the development of *C. elegans* larvae, approximately 100 eggs were incubated on NGM plates containing rCYL at 0.02 mg/mL (3.1 µM) or 0.1 mg/mL (16 µM) (referred to as rCYL–medium) or bovine serum albumin at 0.1 mg/mL (BSA–medium, as a control). After 24, 48, 72, and 96 h, the number and size of the larvae on each medium were recorded (Figure 1a). The hatching percentage, which was determined by calculating [100 × (number of larvae)/(number of eggs)], after 24 h was highly similar (approximately 80–90%) between the rCYL–medium and BSA–medium, indicating that rCYL did not inhibit hatching. After 96 h, approximately 90% of nematodes survived on each medium, indicating that rCYL was not lethal for nematodes. However, rCYL at 0.1 mg/mL in the diet medium apparently reduced the body volume of nematodes to approximately 60% (*p*-value < 0.01) of the body volume of the control (Figure 1a), indicating that rCYL inhibited the development of nematodes. Representative images of the nematodes reared on medium containing 0.1 mg/mL BSA and 0.1 mg/mL rCYL for 96 h are shown in Figure 1b. The nematode reared on BSA–medium was pregnant, i.e., contained eggs in its body, whereas the nematode reared on rCYL–medium had a relatively small body size and was not pregnant. Among the approximately 100 nematodes reared on BSA–medium, 98.6% ± 1.3% (the mean value ± standard deviation) of nematodes were pregnant after 96 h incubation. By contrast, 94.7% ± 0.6% (*p* < 0.01) and 81.0% ± 2.0% (*p* < 0.001) of nematodes reared on medium containing rCYL at 0.02 mg/mL and 0.1 mg/mL, respectively, were pregnant. Thus, rCYL in the diet medium reduced the ratio of pregnant nematodes, dose-dependently.

To further evaluate the effect of rCYL on the egg-laying behavior of nematodes, mature nematodes that were reared on medium containing 0.1 mg/mL rCYL or 0.1 mg/mL BSA were prepared by incubating eggs on each medium for four days, and were defined as the first-generation (G1). One G1 nematode reared on rCYL–medium was isolated, transferred onto fresh rCYL–medium, and incubated for 3 h for egg-laying (*n* = 5). Similarly, one G1 nematode reared on BSA–medium was isolated, transferred onto fresh BSA–medium, and incubated for 3 h (*n* = 5). After removing the G1 nematodes, the G2 (the second-generation) larvae that had developed on each medium were counted after 48 h. One G1 nematode reared on BSA–medium generated 13.6 ± 4.3 G2 larvae on average. By contrast, one G1 nematode reared on rCYL–medium generated only 8.2 ± 3.1 G2 larvae. Thus, the ingestion of 0.1 mg/mL rCYL by G1 nematodes reduced the number of G2 larvae to approximately 60% that of the control. Because rCYL does not interfere with hatching or the survival of larvae, this result indicated that rCYL repressed the egg-laying behavior of G1 nematodes.

### 2.2. Inhibition of the Development and Egg-Laying of Drosophila

To test the effect of rCYL on the development of *Drosophila*, 20 eggs were incubated on medium containing phosphate-buffered saline (PBS, as a control) or rCYL at 0.02 mg/mL (3.1 µM) or 0.1 mg/mL (16 µM). After 2, 9, and 13 days, the larvae, pupae, and adult flies that had developed on each medium were counted (Appendix A). On media containing PBS or rCYL at 0.02 mg/mL, 18.7 ± 1.5 and 19.3 ± 0.6 larvae developed, respectively, whereas on medium containing rCYL at 0.1 mg/mL, 17.3 ± 0.6 larvae developed, indicating that rCYL at 0.1 mg/mL has a marginal effect on hatching. In pupation, media containing PBS or rCYL at 0.02 mg/mL resulted in 17.0 ± 2.6 and 17.3 ± 0.6 pupae, respectively, whereas medium containing rCYL at 0.1 mg/mL resulted in 12.6 ± 3.0 pupae. Thus, rCYL at 0.1 mg/mL repressed pupation to approximately 74% that of the control. Larvae that failed in pupation remained in the medium without movement and appeared dead. During emergence, medium containing PBS or rCYL at 0.02 mg/mL resulted in 16.3 ± 2.5 and 13.0 ± 3.6 adult flies, respectively, whereas media containing rCYL at 0.1 mg/mL resulted in 6.6 ± 3.2 (*p* < 0.05) adult flies. The number of adult flies corresponded to approximately 40% those of the control. These results indicated that rCYL at 0.1 mg/mL substantially inhibits the emergence of *Drosophila*.

Next, the effects of rCYL ingested by adult flies on their egg-laying behavior and the development of the next generation were examined (Figure 2). Within 12 h of emergence, virgin males and females of the G1 generation (first-generation) were separately cultivated on BSA–medium (BSA at 0.3 mg/mL) or rCYL–medium (rCYL at 0.3 mg/mL, 47 µM) for 53 h, as shown in Figure 2a. Evasive action in response to rCYL was not observed in the adult flies. Next, 10 G1 males and 10 G1 females reared on rCYL–medium or BSA–medium were randomly selected, transferred onto fresh BSA–medium, and mixed in the following combinations, “BSA–male/BSA–female”, “rCYL–male/BSA–female”, “BSA–male/rCYL–female”, and “rCYL–male/rCYL–female” (Figure 2b), in which “BSA–male” or “rCYL–female”, for example, indicates a G1 male reared on BSA–medium or a G1 female reared on rCYL–medium, respectively. In addition, 10 G1 males and 10 G1 females in the combination “rCYL–male/rCYL–female” were also transferred onto fresh rCYL–medium (rCYL at 0.3 mg/mL) (Figure 2b). Thus, five different combinations of G1 adult flies, namely “BSA–male/BSA–female/BSA–medium” (combination I), “rCYL–male/BSA–female/BSA–medium” (II), “BSA–male/rCYL–female/BSA–medium” (III), “rCYL–male/rCYL–female/BSA–medium” (IV), and “rCYL–male/rCYL–female/rCYL–medium” (V), were prepared. Then, the G1 adult flies were allowed to mate freely and lay eggs on each medium for 62 h (Figure 2b). During this period, the G1 adult flies in all combinations survived, indicating that rCYL at 0.3 mg/mL was not lethal for adult flies. After removing the G1 flies from the medium, the G2 (second-generation) eggs laid on each medium were counted (Figure 2c). The ingestion of rCYL at 0.3 mg/mL in the combinations (II)–(V) reduced the number of G2 eggs to 66–72% that of the control (I), indicating that rCYL inhibited the egg-laying of G1 adult flies.

After 7 and 10 days, the total G2 pupae and G2 adult flies that had developed on each medium were counted (Figure 2d,e). The combinations “rCYL–male/BSA–female/BSA–medium” (II) and “BSA–male/rCYL–female/BSA–medium” (III) reduced the numbers of G2 pupae to 64% and 26% that of the control “BSA–male/BSA–female/BSA–medium” (I), respectively (Figure 2d). Interestingly, the ingestion of rCYL by G1 female flies more severely reduced the number of G2 pupae compared with G1 male flies (Figure 2d). The combination “rCYL–male/rCYL–female/BSA–medium” (IV) resulted in a greater reduction in the number of G2 pupae, corresponding to only 10% that of the control (I) (Figure 2d). This result indicated that following the ingestion of rCYL by both G1 male and female flies, a synergic defect was observed in G2 pupation. The combination “rCYL–male/rCYL–female/rCYL–medium” (V) generated no pupae (Figure 2d). This seemed to be due to the longer exposure of both G1 male and female adults to rCYL [115 h in total (Figure 2a,b), compared with 53 h (Figure 2a) for combination (IV)], in addition to the ingestion of rCYL in the medium by G2 larvae for 7 days (Figure 2d). Regarding G2 emergence (Figure 2e), the combinations “rCYL–male/BSA–female/BSA–medium” (II), “BSA–male/rCYL–female/BSA–medium” (III), and “rCYL–male/rCYL–female/BSA–medium” (IV) reduced the numbers of G2 adult flies to 48%, 23%, and 9% those of the control (I), respectively. Thus, the efficiencies of G2 emergence were further decreased in the combinations (II)–(IV), indicating that rCYL repressed the emergence of G2 adult flies. Sexual bias between the numbers of G2 male and female adult flies in the combinations (II)–(IV) was not observed.

After removing the G1 adult flies from each medium in the combinations (I)–(V), they were transferred onto fresh BSA–medium (Figure 2f). Their combinations were numbered afresh as (1)–(5), because all combinations were placed on BSA–medium. Then, the G1 adult flies were allowed to mate freely and lay eggs (referred to as G2-2 eggs) for 3 days on BSA–medium. During this period, the G1 adult flies survived and were then removed from each medium (Figure 2g). After 9 days, the G2-2 adult flies developed from the G2-2 eggs were counted for each combination (Figure 2h). Compared with the number of G2 adult flies in Figure 2e, the numbers of G2-2 adult flies in Figure 2h were largely restored in the combinations (2)–(4). This result indicated that the fecundity of the G1 adult flies was gradually alleviated by ceasing the ingestion of rCYL. However, the number of G2-2 adult flies in combination (5) remained at only 32% that of the control (Figure 2h). This result confirmed that the longer exposure times of both G1 male and female flies to rCYL caused more severe defects in egg-laying and the development of the next generation.

The G2-2 adult flies in the five combinations (1)–(5) were further subjected to free-mating and egg-laying experiments (Figure 2i,j). Ten randomly selected G2-2 male and female adult flies of the combinations (1)–(5) were transferred onto fresh BSA–medium (Figure 2i) to generate G3 (third-generation) adult flies (Figure 2k). The number of G3 adult flies in combination (5) corresponded to 81% that of the control (Figure 2k), which was lower than the control despite the ingestion of BSA–medium by the G2-2 adult flies. This result indicated that the long period of ingestion of rCYL (115 h in total) by the G1 adult flies may have led to certain defects in the egg-laying of G2-2 adult flies and the development of G3 larvae.

### 2.3. Detection of Fluorescently Labeled rCYL in Nematodes and Drosophila

In an attempt to identify the target sites of rCYL in *C. elegans* and *D. melanogaster*, rCYL was labeled with fluorescein isothiocyanate (FITC), referred to here as FITC-rCYL. FITC-rCYL exhibited the same antifungal activity as that of rCYL against *Aspergillus nidulans* [27]. FITC-labeled BSA (FITC-BSA), as a control, was commercially obtained. Figure 3a shows that the nematode fed on medium containing 0.2 mg/mL FITC-BSA exhibited a faint fluorescent signal along the intestine. Figure 3b shows that the nematode fed on medium containing 0.2 mg/mL (31 µM) FITC-rCYL exhibited intense fluorescent signals along the pharynx “https://www.wormatlas.org/hermaphrodite/pharynx/mainframe.htm (accessed on 26 February 2025)”, in addition to a faint signal along the intestine. The pharynx is a muscular feeding organ that begins at the buccal cavity and terminates posteriorly at the terminal bulb (Figure 3c) [31]. Pharyngeal contractions cause the posterior movement of alimentary contents towards the intestine, and the grinder ruptures the contents during feeding [32]. The pharynx is lined with cuticle (Figure 3c) [33,34]. Although the chemical composition of the pharyngeal cuticle remains poorly defined, it contains a chitin–chitosan matrix, proteoglycans, and amyloidogenic proteins [34,35,36,37,38]. Figure 3d,e show the staining pattern of Calcofluor white (CFW), which has been used as a chitin probe in many biological systems and was reported to specifically stain the pharyngeal cuticle in nematodes [38]. Figure 3f,g show that FITC-rCYL localized at the pharyngeal cuticle of the nematode, which was stained with CFW.

Figure 4a,b show the stereo-microscopic and fluorescence images, respectively, of the female adult fly that was reared on medium containing 0.3 mg/mL FITC-BSA and dissected as a control. Figure 4c–h shows the stereo-microscopic and fluorescence images of the female adult flies that were reared on medium containing 0.3 mg/mL (47 µM) FITC-rCYL and then dissected. The females fed on FITC-BSA did not show an apparent fluorescent signal (Figure 4b), whereas the females fed on FITC-rCYL showed an intense fluorescent signal at the midgut and a weak signal at the foregut, hindgut, crop, ovary, and malpighian tubules (Figure 4d,f–h) [39,40]. The image of the ovary in Figure 4f is expanded in Figure 4g,h, revealing that unfertilized eggs also emitted a fluorescent signal. Figure 4i,j show images of a male adult fly that was reared on medium containing 0.3 mg/mL FITC-BSA. Figure 4k–o show images of male adult flies reared on medium containing 0.3 mg/mL FITC-rCYL. In males, an intense fluorescent signal was detected at the midgut and a weak signal at the foregut, hindgut, crop, testis, and malpighian tubules (Figure 4l,m,o). The image of the midgut in Figure 4l is expanded in Figure 4m, revealing fluorescent particles in the midgut. Figure 4p–u show the eggs laid by the flies that were fed on 0.3 mg/mL FITC-BSA and 0.3 mg/mL FITC-rCYL, respectively. The eggs laid by the adult flies fed on FITC-rCYL showed weak but detectable fluorescent signals in an unevenly distributed pattern (Figure 4s), compared with the eggs of adult flies fed on FITC-BSA (Figure 4q). Some eggs of adult flies fed on FITC-rCYL showed multiple bands around the center of the eggs (Figure 4u), which became apparent when the fluorescence detection sensitivity was increased. Figure 4v,w show the larvae that hatched from the eggs of adult flies fed on medium containing 0.3 mg/mL FITC-BSA and then grown on FITC–BSA–medium. Figure 4x–z shows the larvae that hatched from the eggs of adult flies fed on medium containing 0.3 mg/mL FITC-rCYL and then grown on FITC–rCYL–medium. Compared with the control (Figure 4w), the larvae hatched from the eggs of adult flies fed on FITC-rCYL and reared on FITC–rCYL–medium (Figure 4y,z) showed an intense fluorescent signal at the midgut and weak fluorescent signals at the foregut, hindgut, and malpighian tubules.

Figure 5 shows fluorescence microscopic images of the midguts of adult flies and larvae that were fed on medium containing 0.3 mg/mL FITC-BSA or 0.3 mg/mL FITC-rCYL, dissected, and fixed prior to observation. In contrast to the midgut of the adult fly fed on FITC-BSA (Figure 5a), the midguts of adult flies fed on FITC-rCYL showed two intense fluorescent layers, one representing the midgut tissue and the other representing the alimentary contents (Figure 5b,c). Similarly, two fluorescent layers were also observed in the midgut of larvae that ingested FITC-rCYL (Figure 5d). The region indicated by a broken line in Figure 5d was expanded and brightened in Figure 5e to clearly reveal the midgut tissue and the fluorescent contents. One part of the midgut tissue was broken down during sample preparation, and the midgut contents were bare and observed to be composed of many fluorescent particles (Figure 5e). Figure 5f shows the midgut of an adult fly that was fed FITC-rCYL and then fasted in PBS for 6 h. The fluorescent contents were not observed. Because foods normally travel the entire length of the digestive tract in less than 1 h [41], the fluorescent contents would have been excreted from the digestive tract during starvation.

### 2.4. Stability of rCYL Against Proteolytic Enzymes In Vitro

Following the observation of the fluorescent contents in the midgut of *Drosophila*, the stability of rCYL against proteolytic enzymes was tested in vitro. rCYL was incubated with bovine trypsin, α-chymotrypsin, and porcine pepsin, and the reaction mixtures were resolved by SDS-PAGE (Figure 6a, left). The results indicated that rCYL was stable against the proteolytic enzymes tested. By contrast, rCYL, which was first reduced and denatured by treatment with dithiothreitol (DTT) at 100 °C, was completely hydrolyzed by the proteases (Figure 6a, right), indicating that multiple disulfide bridges in rCYL were essential for stability against proteolytic enzymes.

The observation of the fluorescent contents in the midgut of *Drosophila* also raised the possibility that rCYL might inhibit the activities of digestive enzymes. Some entomotoxic proteins have been reported to show inhibitory activity against proteolytic enzymes in vitro [18,42]. To test this hypothesis, rCYL was preincubated with bovine trypsin and then mixed with a trypsin-specific substrate, Bz-L-Arg-pNA (benzoyl-L-arginyl para-nitroanilide), and the proteolytic reaction was monitored calorimetrically at 405 nm for liberated pNA (Figure 6b). The result indicated that rCYL did not inhibit proteolysis of the substrate by trypsin. A similar experiment with bovine α-chymotrypsin was also performed using a specific substrate, Bz-L-Tyr-pNA (benzoyl-L-tyrosyl para-nitroanilide). The result indicated that rCYL did not inhibit α-chymotrypsin.

Glycolytic enzymes play important roles in the digestion of plant starch in the insect gut. Some insecticidal plant defensins, including VrD1 from *Vigna radiata* [15], VuD1 from *Vigna unguiculata* [16], and ZmDEF1 from *Zea mays* [43], have been reported to inhibit insect α-amylases in vitro. Whether rCYL shows inhibitory activity against glycolytic enzymes, α-amylase from *Bacillus amyloliquefaciens,* and glucoamylase from *Aspergillus niger*, was tested using the Bernfeld method [44]. As shown in Figure 6c,d, rCYL did not inhibit the α-amylase or glucoamylase.

**Figure 6 toxins-17-00118-f006:**
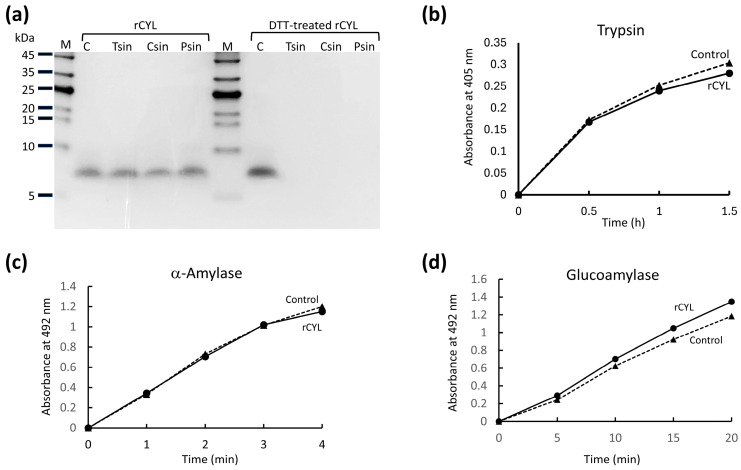
rCYL is stable against proteolytic enzymes but does not inhibit proteolytic and glycolytic enzymes. (**a**) rCYL was incubated with bovine trypsin (Tsin), α-chymotrypsin (Csin), and porcine pepsin (Psin), and the reaction mixtures were resolved by SDS-PAGE. As a control (C), rCYL was incubated in a buffer without enzymes. rCYL was stable against proteolytic reactions by the enzymes (*n* = 2). By contrast, rCYL, which was first reduced and denatured by treatment with dithiothreitol at 100 °C (DTT-treated rCYL), was completely hydrolyzed by the proteases (*n* = 2). (**b**) rCYL was preincubated with bovine trypsin and then mixed with a trypsin-specific substrate Bz-L-Arg-pNA, and the proteolytic reaction was monitored calorimetrically for liberated pNA. rCYL did not inhibit proteolysis of the substrate by trypsin (*n* = 3). (**c**,**d**) α-Amylase from *Bacillus amyloliquefaciens* or glucoamylase from *Aspergillus niger* were preincubated with rCYL or the same amount of buffer (Control), and then mixed with starch. The liberated oligosaccharides and glucose were detected using the Bernfeld method [44]. rCYL did not inhibit these glycolytic enzymes (*n* = 3).

### 2.5. Protein Cross-Linking of rCYL

In the previous immunohistochemical analysis of the fruiting body of *C. cylindracea*, we observed that native cylindracin exists in the outermost layer of the pileus surface, with a dot-like staining pattern, and this result raised the possibility that cylindracin might exist in homopolymeric complexes [27]. In this study, the midguts of the adult flies and larvae fed on FITC-rCYL were observed to contain alimentary contents composed of many fluorescent particles (Figure 5). Although we do not have evidence that shows how the fluorescent particles formed, the intense fluorescence signal of the particles suggested that many FITC-rCYL molecules gathered in the particles. Some plant defensins, including NaD1, have been reported to form dimers and higher oligomers in vitro, and that dimer formation is critically involved in their antibiotic activities [45]. A fungal defensin plectasin variant was reported to assemble into fibrils, in a pH- and concentration-dependent manner [46]. Legume lectins also form homodimers or homotetramers [19]. To test the self-assembly competence of rCYL in vitro, protein cross-linking followed by SDS-PAGE was performed (Figure 7). The cross-linking reagent used here, BS(PEG)_9_, contains *N*-hydroxysuccinimide esters at both ends of the nonaethylene glycol–spacer arm [47]. The *N*-hydroxysuccinimide esters react with free α-amino and *ε*-amino groups of proteins, which cross-link proteins through the formation of covalent amide bonds. As shown in Figure 7a, rCYL migrated as a main band at ~7 kDa (monomer) in the absence of the BS(PEG)_9_ cross-linker. The minor bands close to the main band likely formed via partial disulfide formation during electrophoresis under the conditions used here, although rCYL was reduced and denatured by treatment with DTT and SDS prior to SDS-PAGE. rCYL at 0.25 mg/mL (39 µM) was incubated with 0.005–5 mM BS(PEG)_9_. As the concentration of BS(PEG)_9_ was increased from 5 to 500 µM, an additional band appeared at a relative molecular weight of ~13 kDa that corresponded to the molecular weight of an rCYL dimer (Figure 7a). Interestingly, when the cross-linker was employed at 5 mM, the protein bands virtually disappeared from the gel. In Figure 7b, as the concentration of rCYL increased from 0.25 mg/mL (39 µM) to 2.0 mg/mL (312 µM) in the presence of BS(PEG)_9_ at 500 µM, another protein band appeared at a relative molecular weight of ~24 kDa that corresponded to a rCYL tetramer. These results indicated that rCYL has the potential to self-assemble into homodimers and homotetramers, at least under the conditions used here.

### 2.6. Interaction of rCYL with Heparin and Heparan Sulfate

Considering the localization of FITC-rCYL at the pharyngeal cuticle of nematodes and the midgut of *Drosophila*, rCYL might bind to chitin or glycosaminoglycans, because these polysaccharides are contained in the pharyngeal cuticle of nematodes, the epithelial cell surface of the midgut of *Drosophila*, and the PM that lines the midgut lumen [20,34,35,36,48,49]. To test the binding to chitin, rCYL was incubated with chitin powder in 20 mM sodium phosphate buffer and centrifuged prior to SDS-PAGE analysis. rCYL was observed in the supernatant fraction but not in the chitin pellet, indicating that rCYL does not bind to chitin.

Glycosaminoglycans are polysaccharides that are covalently attached to core proteins to form proteoglycans [50,51]. Heparin and heparan sulfate (HS) are members of the glycosaminoglycans, and are highly sulfated linear polysaccharides [50,51]. To test the binding to heparin and HS, rCYL at 0.25 mg/mL (39 µM) was incubated with heparin (from porcine small intestine) or HS (from bovine kidney) in 20 mM sodium phosphate (pH 7.0) and 150 mM NaCl, and the reaction mixture was resolved by SDS-PAGE (Appendix A). As the concentration of heparin increased from 0.04 mg/mL to 3.75 mg/mL, the density of the Coomassie brilliant blue-stained band of rCYL at ~7 kDa decreased gradually down to 15% of the control (Appendix A). This result indicated that rCYL bound to heparin to form rCYL–heparin complexes with a high molecular weight. The high-molecular-weight complexes appeared to be stable against heat-treatment in the SDS-PAGE sample buffer containing DTT and SDS, thereby preventing the complexes from being electrophoresed in the gel (Appendix A). Following the incubation of rCYL with HS at 4.0 mg/mL, the density of the rCYL band decreased to 68% that of the control (Appendix A), indicating that rCYL bound to HS to form high-molecular-weight complexes too. By contrast, SP–Sepharose and chitin did not cause such a reduction in the density of the rCYL band (Appendix A). The view that rCYL forms high-molecular-weight complexes with heparin or HS was supported by the observation of protein bands around the boundary between the stacking gel and running gel of SDS-PAGE (Appendix A). The result of size-exclusion column chromatography using a resin of molecular weight cut-off of 7 kDa also supported this view (Appendix A). rCYL (0.5 mg/mL) incubated with heparin (4 mg/mL) in 20 mM sodium phosphate buffer (pH 7.0) entirely passed through the column without retardation, whereas rCYL itself (0.5 mg/mL) was partly (about 40%) absorbed to the column, and residual 60% of rCYL passed through the column, probably due to dimer formation, as shown in Figure 7.

To evaluate the intensity of the binding to heparin, FITC-rCYL at 0.25 mg/mL was incubated with porcine heparin agarose resin in 20 mM sodium phosphate buffer (Figure 8a). After washing the resin, the resin was incubated in the following different solutions: 20 mM sodium phosphate buffer, 20 mM sodium phosphate containing 0.5 M NaCl, 20 mM sodium phosphate containing 1 M NaCl, 20 mM sodium phosphate containing 1 M NaCl and 4 M urea, or 20 mM sodium phosphate containing 1 M NaCl and 8 M urea. After washing the resin, the fluorescence intensity of the resin was measured (Figure 8a). Following incubation in 20 mM sodium phosphate containing 0.5 M NaCl, the fluorescence intensity of the resin was reduced to approximately half of the initial intensity. An increase in ionic strength from 0.5 M to 1 M NaCl and the addition of a denaturant, 4 M or 8 M urea, did not cause a further reduction in the fluorescence intensity of the resin (Figure 8a). These results indicated that rCYL bound to heparin partly via an ionic interaction, but also by interactions that were stable against a high ionic strength of 1 M NaCl and a high concentration of denaturant, 8 M urea. Figure 8b–e shows the fluorescence images of the resins after treatment in 20 mM sodium phosphate buffer, 20 mM sodium phosphate containing 0.5 M NaCl, 20 mM sodium phosphate containing 1 M NaCl, or 20 mM sodium phosphate containing 1 M NaCl and 8 M urea, respectively. Fluorescence microscopy showed that FITC-rCYL still bound to the resin even after treatment with 1 M NaCl and 8 M urea.

Figure 8f–i show the fluorescence images of the aggregates that were formed in vitro by the incubation of rCYL at 0.4 mg/mL with heparin and HS at 4 mg/mL, respectively, followed by staining with thioflavin T. rCYL alone did not form such fluorescent aggregates. These results indicated that heparin and HS promote the amyloid aggregation of rCYL.

## 3. Discussion

This study showed that orally ingested rCYL inhibits the development of *C. elegans* larvae and reduces the population of pregnant adults, which results in a reduction in the average number of larvae born from one nematode. This study also showed that orally ingested rCYL inhibits the development of larvae in *D. melanogaster*, especially the pupation and emergence steps. Furthermore, the ingestion of rCYL by adult flies reduced their egg-laying frequencies, and repressed the pupation and emergence of the next generation. Interestingly, compared with the ingestion of rCYL by male adult flies, the ingestion of rCYL by female adult flies caused more severe defects in the pupation and emergence of the next generation. The inhibitory effects of rCYL gradually disappeared after ceasing the ingestion of rCYL by adult flies. Thus, rCYL has a unique function in controlling egg-laying and the development of the next generation in both nematodes and *Drosophila*.

Using FITC-labeled rCYL, we observed that ingested rCYL accumulates at the pharyngeal cuticle of nematodes. The pharyngeal cuticle is composed of a chitin–chitosan matrix, proteoglycans, and amyloidogenic proteins [34,35,36,37,38]. A polysaccharide-binding assay using SDS-PAGE indicated that rCYL binds to, and forms, a high-molecular-weight complex with heparin and HS, but not with chitin. We also observed, with the aid of an amyloid-specific dye thioflavin T, that rCYL forms amyloid aggregates in the presence of heparin and HS. Heparin and HS are glycosaminoglycans that possess a large number of sulfate groups along linear polysaccharide chains comprising repeating disaccharides, uronic acid and hexosamines [50,51]. In mammals, heparin is produced exclusively in connective tissue-type mast cells, usually existing as free polysaccharides after enzymatic cleavage from core proteins [50]. HS is produced by most cells in the body, is generally less sulfated than heparin in mammals, and exists largely in proteoglycan form at cell surfaces and in the extracellular matrix [50,51]. Heparin and HS have been reported to promote the formation of amyloid aggregates or fibrils of various proteins, including prion protein [52], apolipoprotein [53], and β2-microgloblin [54]. In humans, HS proteoglycans are ubiquitous components of pathologic amyloid deposits in the organs of patients with Alzheimer’s disease, rheumatoid arthritis, and diabetes [55,56,57]. The results obtained here raise the hypothesis that rCYL might bind to the glycosaminoglycans of proteoglycans constituting the pharyngeal cuticle of the nematode, and interfere with the function of the pharyngeal cuticle such as posterior movement and the disintegration of alimentary contents, which results in the malnutrition of nematodes and repression of the nematode’s development [34]. In addition, the aggregates of rCYL deposited on the pharyngeal cuticle might also inhibit the construction and deconstruction of the pharyngeal cuticle during molting, which would also interfere with the development of nematodes [58].

We observed an intense fluorescent signal following the ingestion of FITC-rCYL at the midgut of larvae and adult flies of *Drosophila*. The midgut is the primary site in the intestine for digestion and nutrient absorption [59]. The fluorescence images of the midgut showed two layers: one was the midgut tissue and the other was the alimentary contents in the lumen. The contents of the midgut lumen were composed of many fluorescent particles. Our in vitro experiments showed that rCYL does not inhibit proteolytic trypsin and α-chymotrypsin, or glycolytic α-amylase and glucoamylase. Instead, rCYL was stable against trypsin, α-chymotrypsin, and pepsin. Therefore, the fluorescent particles in the midgut contents likely contain undigested FITC-rCYL. Considering the binding and amyloid formation of rCYL with heparin and HS, as observed here, the ingested FITC-rCYL might bind to the glycosaminoglycans of proteoglycans on the epithelial cell surface and/or in the PM of the midgut, thereby aggregating to form fluorescent particles.

The PM is ~300 nm thick and is composed of a three-dimensional meshwork of chitin fibers that are cross-linked by chitin-binding proteins, including proteoglycans. The PM provides an essential line of defense for epithelial cells from direct physical contact with food particles, as well as playing a crucial role in digestion and nutrient absorption by acting as a molecular sieve [20,48,49,60]. Lectins that bind to the PM, and the antibodies or chemicals that bind to chitin in the PM, can disrupt its permeability, thereby affecting insect growth, especially during the larval stages [19,61,62,63,64]. The binding and aggregation of rCYL on the glycosaminoglycans of proteoglycans in the PM could interfere with the permeability and the absorption of nutrients, which would inhibit the development of *Drosophila* larvae [65].

rCYL has a relatively small molecular weight (6.4 kDa), and may therefore pass through the PM and reach the midgut epithelial cell surface, as observed for some lectins [66]. If this occurs, rCYL might bind to the glycosaminoglycans of proteoglycans on the midgut epithelial cell surface and aggregates, which might interfere with important functions of the midgut cells, such as the secretion of digestive enzymes and the absorption of nutrients. In fact, some lectins exert insecticidal activity through binding to glycoproteins in the midgut epithelial cells [19,67]. In this study, rCYL showed less interaction with HS (from bovine kidney) than with heparin (from porcine small intestine); this difference could reflect lower affinity due to the number and/or position of sulfate groups in HS and heparin [50,51]. Because the HS of *Drosophila* contains more sulfate groups than the HS of mammals [68], rCYL might bind to HS in the midgut of *Drosophila* with higher affinity than to the HS used in this study. Thus, the binding of rCYL to glycosaminoglycans and aggregate formation in the PM and/or the midgut epithelial cells would cause malnutrition and result in the inhibition of the development of *Drosophila*.

In many species, including *Drosophila*, the nutritional condition of the female is closely related to egg production. Malnutrition represses egg production and reduces the size and number of eggs [40]. If rCYL causes malnutrition in female adult flies, it would lead to the suppression of egg-laying, as observed here. However, it remains unknown whether the malnutrition of male adult flies also suppresses egg-laying [69], because in this study, the ingestion of rCYL by male adult flies also resulted in suppression of egg-laying. It is also unknown whether the malnutrition of parent adult flies inhibited the development of the next generation, especially the pupation and emergence steps, as the inhibition of both steps was observed in this study. In the female and male adult flies fed on FITC-rCYL, the ovaries, unfertilized eggs, and testes emitted weak but substantial fluorescence, indicating that the ingested FITC-rCYL reached the reproductive organs. Eggs laid by the FITC-rCYL-treated adult flies also showed weak fluorescence. These results indicated that some of the ingested FITC-rCYL crossed the midgut, entered into the hemolymph, and then reached the reproductive organs. This was supported by the fact that the malpighian tubules also emitted fluorescence. Malpighian tubules comprise a single epithelial layer, and play a key role as the main excretory organ by taking up and filtering catabolites, solutes, and water from the surrounding hemolymph, generating the so-called primary urine [70,71]. Some insecticidal lectins cross the insect midgut epithelium and enter the hemolymph [19,67]. Although the complete mechanism by which lectins cross midgut epithelial cells is not yet fully understood, the available evidence implicates clathrin-mediated endocytosis [72,73]. Regarding the cellular uptake of amyloids in the progression of neurodegenerative diseases, HS proteoglycans on the surface of neuronal cells play a critical role in the binding of extracellular aggregates of the Aβ protein, prion protein, or tau protein, and the uptake of aggregates into neuronal cells to seed the further aggregation of normal intracellular proteins [74,75]. As a hypothesis, rCYL might bind to and use a certain receptor on the surface of midgut epithelial cells to cross the midgut and then reach the reproductive organs. It also remains elusive how the binding of rCYL to the reproductive organs occurs and how it affects the development of the next generation of *Drosophila*. In future studies, the identification of physiological target molecules of rCYL in *Drosophila,* as well as in nematodes, and the characterization of the binding mechanisms and intensities would be required. Finally, in this study, biotoxic assays of rCYL were performed using rCYL at a specified concentration, in a contained area (i.e., in a cylindrical glass container or a dish). rCYL might interfere with the growth of beneficial insects that are predatory to pest insects that have consumed rCYL. However, large quantities of intoxicated pest insects would likely need to be consumed to accumulate biologically meaningful doses within beneficial insects. To further clarify the impact of rCYL against various predators, assay systems that can be performed under near-natural conditions are required.

## 4. Conclusions

This study revealed that the antifungal protein cylindracin has the potential to control the egg-laying and development of *D. melanogaster* and *C. elegans*. As far as we know, this is the first report to show that a mushroom protein ingested by adult flies represses the pupation and emergence of the next generation. The inhibitory effects of rCYL are reversible because ceasing the ingestion of rCYL restored the adult flies’ egg-laying ability and the development of the next generation’s larvae. This point is important because rCYL would not damage the environment unnecessarily. The specific accumulation of rCYL at the pharyngeal cuticle of nematodes and the midgut of *Drosophila* has a negative effect on digestion and nutrient absorption, causing malnutrition, which ultimately results in defects in larval development and egg-laying frequencies. However, it remains to be determined how rCYL interferes with the development of the next generation of *Drosophila*. Although the toxicity observed here for nematodes and *Drosophila* seemingly resembles, in part, the toxicities caused by some lectins, rCYL is distinct from lectins in its specific protein structure, which includes multiple disulfide bridges. Further work is required to understand the molecular mechanism underlying the toxic effects of rCYL, which may pave the way to developing a biomaterial to control pest development and egg-laying.

## 5. Materials and Methods

### 5.1. Animals

Wild-type *Caenorhabditis elegans* strain N2 was obtained from the Caenorhabditis Genetics Center (Minneapolis, MN, USA) and grown at 20 °C on standard nematode growth medium (NGM) plates [0.25% (*w*/*v*) peptone, 0.3% (*w*/*v*) NaCl, 1.7% (*w*/*v*) agar, 5 µg/mL cholesterol, and 25 mM potassium phosphate pH 6, 1 mM MgSO_4_, 1 mM CaCl_2_] seeded with *Escherichia coli* OP50 [76]. Synchronized eggs were obtained by the alkali-bleaching method, according to the WormBook guidelines “http://www.wormbook.org/ (accessed on 27 Februsary 2025)”, as described previously [77].

*Drosophila melanogaster* Canton-S strain was reared at 25 °C on standard agar medium [0.8% (*w*/*v*) agar, 3% (*w*/*v*) yeast, 5% (*w*/*v*) glucose, 4% (*w*/*v*) cornmeal, 0.015% (*w*/*v*) propionic acid].

### 5.2. Preparation of rCYL

Recombinant protein representing the Cys-rich domain (amino acid residues from Ala-37 to Ala-95) of cylindracin, possessing the *N*-terminal extra sequence (EFHHHHHHGEVELEVLFQGP) including a 6 × His-tag sequence (underlined) followed by the HRV-3C protease recognition sequence (double underlined), was expressed using *Pichia pastoris* SMD1168H cells that were transformed with the plasmids pPICZαA-6 × His-Cc-PRI3(37–95), as described previously [27]. Briefly, the *Pichia* cells were cultivated in YPM complex medium [100 mM potassium phosphate (pH 6.0), 1% yeast extract, 2% peptone, 0.5% methanol, 1.34% yeast nitrogen base without amino acids, 4 × 10^−5^% biotin] at 30 °C for 48 h. After centrifugation, the recombinant protein in the supernatant was precipitated by salting out using ammonium sulfate and then purified by a HisPur Cobalt Superflow Agarose column (Thermo Fisher Scientific, Waltham, MA, USA). The obtained recombinant protein was incubated with HRV-3C protease (Japan Bioserum, Hiroshima, Japan) to remove the *N*-terminal His-tag sequence, as described previously [27]. The resultant recombinant Cys-rich domain of cylindracin (rCYL), containing the extra Gly–Pro sequence at the *N*-terminus, was purified using SP–Sepharose resin (Cytiva, Amersham, UK) in 20 mM HEPES-NaOH, pH 7.0, and eluted with the same buffer containing 0.5 M NaCl. rCYL was finally purified by reverse-phase high-performance liquid chromatography (RP-HPLC). The protein concentration was determined spectrophotometrically using a molar extinction coefficient (*ε*) of 4170 M^−1^ cm^−1^ at 280 nm.

### 5.3. Fluorescent Labeling of rCYL

rCYL in 20 mM sodium phosphate, pH 7.0, was mixed with 0.25 mL of 0.1 M sodium bicarbonate buffer, pH 9.5, and then water was added to a final volume of 0.5 mL. To this solution, 1 mg/mL of FITC (fluorescein isothiocyanate, *ε* = 68,000 at 494 nm) (Dojindo Laboratories, Kumamoto, Japan) in acetone was added at a molar ratio of 0.1–0.5 to the protein. The solution was stirred in the dark at room temperature for 2 h, and then dialyzed against 20 mM sodium phosphate, pH 7.0, using a Slide-A-lyzer cassette (Thermo Fisher Scientific). The labeling ratio was calculated to be ~10–20% by dividing the FITC concentration by the protein concentration in the solution. The FITC-labeled cylindracin (FITC-rCYL) retained antifungal activity equal to that of the unlabeled protein against *A. nidulans*. FITC-labeled bovine serum albumin (FITC-BSA, Sigma-Aldrich, St. Louis, MO, USA) was mixed with non-labeled BSA (Sigma-Aldrich) to adjust the FITC labeling ratio to ~20%.

### 5.4. Biotoxicity Assay Against C. elegans

To examine the effect of rCYL on the development of nematodes, approximately 100 synchronized eggs were placed on an NGM plate seeded with *E. coli* OP50, containing 0.02 mg/mL rCYL or 0.1 mg/mL rCYL (referred to as rCYL–medium), or 0.1 mg/mL bovine serum albumin (BSA–medium) as a control, and incubated at 20 °C, in triplicate per condition. After 24, 48, 72, and 96 h, the numbers and the body sizes of the nematodes developed on each medium were recorded (*n* = 6). The total body volumes and body lengths of the nematodes were measured, as described previously [30], using an automated image-analyzing device ‘Senchu-gazou-kaiseki-souchi SVK-3A’ (Showa Electric Laboratory Co., Ltd., Fukuoka, Japan). The ratio of pregnant nematodes to the total number of nematodes reared on each medium was determined in triplicate per condition after 96 h incubation.

To examine the effect on the generation shift, approximately 100 eggs were placed on rCYL–medium (0.1 mg/mL rCYL) or BSA–medium (0.1 mg/mL BSA), and incubated at 20 °C for 96 h to generate mature nematodes of the G1 generation. One mature G1 nematode was transferred onto fresh rCYL–medium or BSA–medium, cultivated continuously at 20 °C for 3 h, and then removed from the medium. The eggs laid on the rCYL–medium or BSA–medium were further incubated at 20 °C for 48 h, and G2 larvae that developed on each medium were counted (*n* = 5).

### 5.5. Biotoxicity Assay Against D. melanogaster

One or two milliliters of solid rearing medium containing rCYL (final concentration: 0.02 or 0.1 mg/mL, referred to as rCYL–medium), bovine serum albumin (0.1 mg/mL, BSA–medium, as a control), or phosphate-buffered saline (PBS–medium, as a control) were prepared in a cylindrical glass container (2.5 cm in diameter × 6 cm in height). To test the effect of these media on the development of *Drosophila*, 20 eggs of adult flies reared on standard medium were placed onto the rCYL–medium or the PBS–medium and incubated at 25 °C, in triplicate per condition. The total numbers of larvae, pupae, and adult flies developed on each medium were determined after 2, 9, and 13 days, respectively.

To examine the effects of rCYL on egg-laying and the development of the next generation, male and female adult flies (G1 generation) were separated within 12 h after emergence and reared on solid medium containing 0.3 mg/mL rCYL or 0.3 mg/mL BSA at 25 °C for 53 h, as shown in Figure 2a. Then, 10 each of the virgin G1 males and females reared on the rCYL–medium or the BSA–medium were transferred onto fresh BSA–medium and mixed in the following combinations: “BSA–male/BSA–female”, “rCYL–male/BSA–female”, “BSA–male/rCYL–female”, and “rCYL–male/rCYL–female” (Figure 2b). In addition, 10 each of the G1 males and females in the combination “rCYL–male/rCYL–female” were transferred on to fresh rCYL–medium (Figure 2b). The G1 adult flies in the five combinations, “BSA–male/BSA–female/BSA–medium”, numbered as (I) “rCYL–male/BSA–female/BSA–medium” (II), “BSA–male/rCYL–female/BSA–medium” (III), “rCYL–male/rCYL–female/BSA–medium” (IV), and “rCYL–male/rCYL–female/rCYL–medium” (V), mated freely and laid eggs on each medium at 25 °C for 62 h, in triplicate per combination (Figure 2b). After removing the G1 adult flies from each medium, the G2 eggs laid on each medium were counted (Figure 2c). After another 7 and 10 days, the total numbers of G2 pupae and G2 adult flies on each medium were counted, respectively (Figure 2d,e).

The removed G1 adult flies of the five combinations (I)–(V) in Figure 2c were transferred onto fresh BSA medium, in triplicate per combination, and their combinations were numbered afresh as (1)–(5) (Figure 2f). The G1 adult flies in each combination mated freely and laid eggs. After removing the G1 adult flies from each medium (Figure 2g), the emerged adult flies (referred to as G2-2 adult flies) for each combination were counted every 3 days for 9 days (Figure 2h).

Similarly, 10 of each of the G2-2 male and female adult flies in combinations (1)–(5) in Figure 2h were transferred onto fresh BSA–medium, in triplicate per combination (Figure 2i), mated freely and laid eggs (Figure 2j). Emerged G3 adult flies were counted every 3 days for 12 days (Figure 2k).

### 5.6. Fluorescence Microscopy of FITC-Labeled rCYL

Adult nematodes were placed on NGM plates seeded with *E. coli* OP50, containing 0.2 mg/mL (31 µM) FITC-rCYL or 0.2 mg/mL FITC-BSA, and incubated at 20 °C for 24 h. The nematodes were put into 100 mM azide and then examined by fluorescence microscopy using an Olympus BX53 microscope (Olympus, Tokyo, Japan) with a U-FBNA miller unit, or a Keyence BZ-X800 microscope (Keyence, Osaka, Japan) with a BZ-X filter GFP.

For the observation of FITC-rCYL in adult flies, virgin male and female adults were separated within 12 h after emergence, and reared on solid medium containing 0.3 mg/mL (47 µM) of FITC-rCYL or 0.3 mg/mL FITC-BSA at 25 °C for 66 h. The male and female adult flies were dissected and observed using a stereo Nikon SMZ1500 microscope (Nikon, Tokyo, Japan) equipped with Micronet i-NTER LENS (Micronet, Kawaguchi, Japan). Some adult flies reared on the medium containing FITC-rCYL were incubated in PBS for 6 h before dissection and examined as 6 h-starved samples. To examine the eggs and larvae, virgin male and female adult flies that were reared separately on medium containing 0.3 mg/mL FITC-rCYL, were transferred onto fresh medium containing 0.3 mg/mL FITC-rCYL, and then mated freely at 25 °C for 24 h. The eggs laid on the medium were collected, manually dechorionated with forceps, and examined using a Nikon SMZ1500 microscope equipped with Micronet i-NTER LENS. The larvae hatched from the eggs were also collected and examined by microscopy. As a control, virgin male and female adult flies that were reared separately on the medium containing 0.3 mg/mL FITC-BSA, were transferred onto fresh medium containing 0.3 mg/mL FITC-BSA, and then mated freely. The eggs laid on the medium containing FITC-BSA and the larvae hatched from the eggs were collected and examined, as detailed above. Fluorescence images of the dissected flies and larvae were also observed using a Keyence BZ-X800 microscope, after fixation with 4% paraformaldehyde, followed by mounting on slide glasses with Vectashield Vibrance Antifade Mounting Medium (Vector Laboratories, Newark, CA, USA).

### 5.7. Stability Test of rCYL Against Proteases

rCYL at a concentration of 0.24 mg/mL in 20 mM Tris-HCl, pH 7.9, was incubated with 0.02 mg/mL bovine trypsin (Sigma-Aldrich) or bovine α-chymotrypsin (Sigma-Aldrich) at 25 °C for 40 min (*n* = 2). To evaluate the significance of the multiple disulfide bridges in the recombinant protein, rCYL in 20 mM Tris-HCl, pH 7.9, was incubated with 10 mM dithiothreitol (DTT) at 100 °C for 3 min, and then an aliquot of the reaction mixture was used in the proteolytic reaction with trypsin or α-chymotrypsin (*n* = 2). rCYL or DTT-treated rCYL were also incubated with 0.02 mg/mL porcine pepsin (Fujifilm Wako Chemicals, Osaka, Japan) in artificial gastric juice (6.2 mg/mL NaCl, 2.2 mg/mL KCl, 0.22 mg/mL CaCl_2_, 1.2 mg/mL NaHCO_3_, pH 2.0) [78] at 25 °C for 40 min (*n* = 2). To detect the proteolytic degradation of the protein, an aliquot of the reaction mixture was applied to sodium dodecyl sulfate-polyacrylamide gel electrophoresis (SDS-PAGE) using a 16% Peptide-PAGE mini gel (TEFCO, Tokyo, Japan) and Tricine electrophoresis buffer (TEFCO).

### 5.8. Protease Inhibition Test of rCYL

The inhibitory activity of rCYL against trypsin or α-chymotrypsin was tested using a colorimetric substrate, as follows. Bovine trypsin (final concentration, 10 µg/mL; Sigma-Aldrich) was preincubated with rCYL (final concentration, 0.1 mg/mL) or the same amount of buffer in 20 mM Tris-HCl, pH 8.0, at 30 °C, mixed with Bz-L-Arg-pNA (benzoyl-L-arginyl para-nitroanilide; final concentration, 0.2 mM; Peptide Institute, Suita, Japan), and incubated for 90 min. Every 30 min, the absorbance of the sample at 405 nm was measured using a Multiskan FC plate reader (Thermo Fisher Scientific) (*n* = 2). Similarly, bovine α-chymotrypsin (final concentration, 10 µg/mL; Sigma-Aldrich) was preincubated with rCYL (final concentration, 0.1 mg/mL) or the same amount of buffer in 20 mM Tris-HCl, pH 8.0, at 30 °C, mixed with Bz-L-Tyr-pNA (benzoyl-L-tyrosyl para-nitroanilide; final concentration, 0.2 mM; Peptide Institute), and incubated for 90 min. Every 30 min, the absorbance of the sample at 405 nm was measured (*n* = 2).


*5.9. α-Amylase and Glucoamylase Inhibition Test of rCYL*


The inhibitory activity of rCYL against α-amylase or glucoamylase was tested using the Bernfeld method [44]. α-Amylase (final concentration, 5 µg/mL) from *Bacillus amyloliquefaciens* (Fujifilm Wako Chemicals) was preincubated with rCYL (final concentration, 0.1 mg/mL) or the same amount of buffer in 20 mM sodium acetate buffer, pH 5, mixed with starch [final concentration, 0.5% (*w*/*v*), Fujifilm Wako Chemicals], and incubated at 26 °C for 4 min. An aliquot of the reaction mixture was mixed with the same amount of 1% (*w*/*v*) 3,5-dinitrosalicilic acid (DNS; Fujifilm Wako Chemicals) solution containing 30% (*w*/*v*) potassium sodium (+)-tartrate 4H_2_O (Fujifilm Wako Chemicals) and 0.4 M NaOH, and then incubated at 100 °C for 5 min. The reaction mixture was five-fold diluted with pure water, and the absorbance of the solution at 492 nm was measured using a Multiskan FC plate reader (Thermo Fisher Scientific) (*n* = 2). Similarly, glucoamylase from *Aspergillus niger* (final concentration: 40 µg/mL, Shin-nihon Kagaku Kogyo, Anjo, Japan) was preincubated with rCYL (final concentration: 0.1 mg/mL) or the same amount of buffer in 20 mM sodium phosphate, pH 6.0, at 26 °C, mixed with starch (final concentration: 0.5%), and then incubated for 20 min. Liberated glucose was detected as detailed above (*n* = 2).

### 5.10. Cross-Linking of rCYL

rCYL at 0.25 mg/mL (39 µM) was cross-linked through primary amino groups by the addition of 0.005–5 mM bis-*N*-succinimidyl-(nonaethylene glycol) ester [BS(PEG)_9_] (Thermo Fisher Scientific) in 20 mM sodium phosphate, pH 7.2, and 150 mM NaCl, at 25 °C for 30 min (*n* = 2). An aliquot of the sample was heat-treated in sample buffer containing 10 mM DTT for SDS-PAGE and resolved by SDS-PAGE using a 16% Peptide-PAGE mini gel (TEFCO) and Tricine electrophoresis buffer (TEFCO). The protein bands were detected using Coomassie brilliant blue staining.

### 5.11. Carbohydrate-Binding Assay

rCYL at 0.25 mg/mL (39 µM) was incubated with heparin (from porcine small intestine, Fujifilm Wako Chemicals), heparan sulfate (from bovine kidney, Sigma-Aldrich), chitin (from crab shell, Fujifilm Wako Chemicals), or SP-Sepharose (Cytiva) at 0.02–4 mg/mL in 20 mM sodium phosphate, pH 7.0, and 150 mM NaCl, at 25 °C for 30 min (*n* = 2). The sample was mixed with sample buffer containing 10 mM DTT and SDS for SDS-PAGE, heated at 100 °C for 3 min, and then resolved by SDS-PAGE using a 16% Peptide-PAGE mini gel (TEFCO) and Tricine electrophoresis buffer (TEFCO). SDS-PAGE was also carried out using hand-made gels composed of 4.5% stacking gel and 18% running gel. The proteins were detected using Coomassie brilliant blue staining or silver staining, and the density of the protein bands was calculated using Image J version 1.54.

rCYL at 0.5 mg/mL was incubated with or without heparin at 4 mg/mL in 100 µL of 20 mM sodium phosphate pH 7.0, at 25 °C for 1 h. Absorbance of the sample solution at 280 nm was recorded prior to size-exclusion chromatography using a Zeba Spin Desalting Columns, 7K MWCO (Thermo Fisher Scientific). After centrifugation, absorbance at 280 nm of the eluted sample solution from the column was recorded. Experiments were performed in triplicate per condition.

Chitin powder (1 mg) was incubated with 20 µg of rCYL in 100 µL of 20 mM sodium phosphate buffer, pH 7.0, at room temperature for 2 h, with gentle agitation (*n* = 2). After centrifugation at 13,000× *g* for 3 min, the supernatant was recovered for SDS-PAGE analysis. The chitin pellet was washed three times with pure water and centrifuged at 13,000× *g* for 3 min each. Then, the chitin pellet was suspended in 100 µL of the sample buffer containing 10 mM DTT, and boiled for 2 min. After centrifugation at 13,000× *g* for 3 min, the supernatant was recovered for SDS-PAGE analysis. SDS-PAGE was performed on 18% polyacrylamide gel.

FITC-rCYL (0.25 mg/mL) was incubated with 100 µL of porcine–heparin agarose resin (M&S TechnoSystems, Osaka, Japan) in 20 mM sodium phosphate buffer, pH 7.0, at room temperature for 1 h. After washing with 20 mM sodium phosphate buffer three times, the resin was incubated with 20 mM sodium phosphate buffer, the same buffer containing 0.5 M NaCl, the same buffer containing 1 M NaCl, the same buffer containing 1 M NaCl plus 4 M urea, or the same buffer containing 1 M NaCl plus 8 M urea at room temperature for 10 min. After washing the resin with 20 mM sodium phosphate buffer, the fluorescence intensity of the resin was measured using a Fluoroskan FL plate reader (Thermo Fisher Scientific) at excitation/emission wavelengths of 485/538 nm, respectively (*n* = 1). The fluorescence of the resin was observed using a Keyence BZ-X800 microscope.

### 5.12. Thioflavin T Fluorescence Assay

rCYL at 0.4 mg/mL (63 µM) was incubated with 4 mg/mL heparin or heparan sulfate in 20 mM phosphate buffer, pH 7.0, containing 150 mM NaCl at 25 °C for 30 min. The reaction mixture was further incubated with 5 µM thioflavin T at 25 °C for 60 min in the dark. An aliquot of the sample was observed using an Olympus BX53 microscope or a Keyence BZ-X800 microscope (*n* = 3).

### 5.13. Statistical Analysis

All experiments subjected to statistical analysis were performed at least in triplicate. The data are represented as the mean values ± SD (standard deviation). Statistical significance was determined using Student’s *t*-test. Significant differences between the control and samples are indicated by *p*-values < 0.05 (*), < 0.01 (**), and < 0.001 (***).

## Figures and Tables

**Figure 1 toxins-17-00118-f001:**
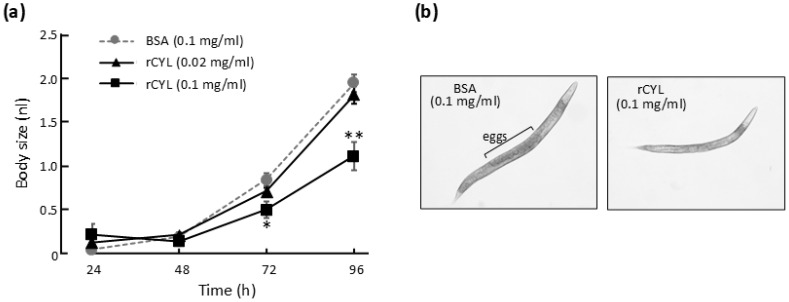
Inhibitory effect of rCYL on the development of nematodes. (**a**) Synchronized eggs were placed on an NGM plate seeded with *Escherichia coli*, containing recombinant cylindracin (rCYL) or bovine serum albumin (BSA) as a control, and incubated in triplicate per condition. After 24, 48, 72, and 96 h, the body volumes (nl) of nematodes on each medium were measured as previously described [30] (*n* = 6). The mean values ± standard deviations (SD) were plotted. Statistically significant differences from the control were determined using Student’s *t*-test and are indicated as * *p* < 0.05, and ** *p* < 0.01. (**b**) Representative images of nematodes reared on medium containing BSA or rCYL after 96 h incubation.

**Figure 2 toxins-17-00118-f002:**
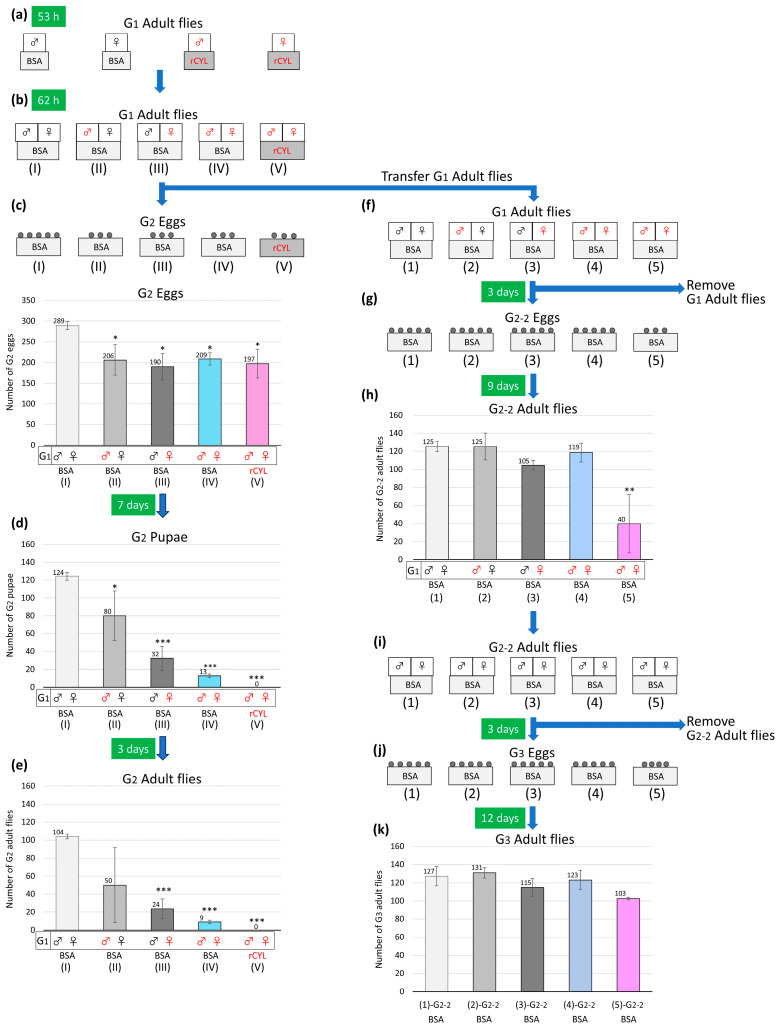
Inhibitory effect of rCYL on the fecundity of *Drosophila*. (**a**) Virgin male and female adult flies of G1 generation were separately cultivated on BSA–medium (BSA at 0.3 mg/mL) or rCYL–medium (rCYL at 0.3 mg/mL) for 53 h. The adult flies reared on rCYL–medium are shown in red. (**b**) Five different combinations (I)–(V) of the virgin G1 male and female adult flies were prepared on fresh BSA–medium or rCYL–medium, and then allowed to mate freely and lay eggs on each medium for 62 h. (**c**) After transferring the G1 adult flies in the combinations (I)–(V) to (**f**), G2 (second-generation) eggs on each medium were counted. The mean value of the G2 eggs is shown on the top of each column. (**d**) After 7 days, G2 pupae on each medium were counted. G2 pupae that formed belatedly were also counted, and the total numbers of the G2 pupae on each medium are shown. (**e**) After 3 days, G2 adult flies emerged from the pupae were counted. (**f**) After transferring from (**c**), the G1 adult flies in the five combinations were allowed to mate freely and lay eggs on fresh BSA–medium. Their combinations were numbered afresh as (1)–(5). (**g**) After 3 days, the G1 adult flies in the five combinations were removed and the eggs (referred to as G2-2 eggs) laid on the BSA–medium were continuously incubated. (**h**) After 9 days, G2-2 adult flies emerged on each medium were counted. (**i**) Ten randomly selected G2-2 male and female adult flies in the combinations (1)–(5) in (**h**) were transferred onto fresh BSA–medium, and allowed to mate freely and lay eggs. (**j**) After 3 days, the G2-2 adult flies were removed. G3 (third-generation) eggs laid on the BSA–medium were continuously incubated. (**k**) After 12 days, G3 adult flies emerged on each medium were counted. All experiments were performed in triplicate per condition, and the mean values ± SD are shown. Statistically significant differences from the control were determined using Student’s *t*-test and are indicated as * *p* < 0.05, ** *p* < 0.01, and *** *p* < 0.001.

**Figure 3 toxins-17-00118-f003:**
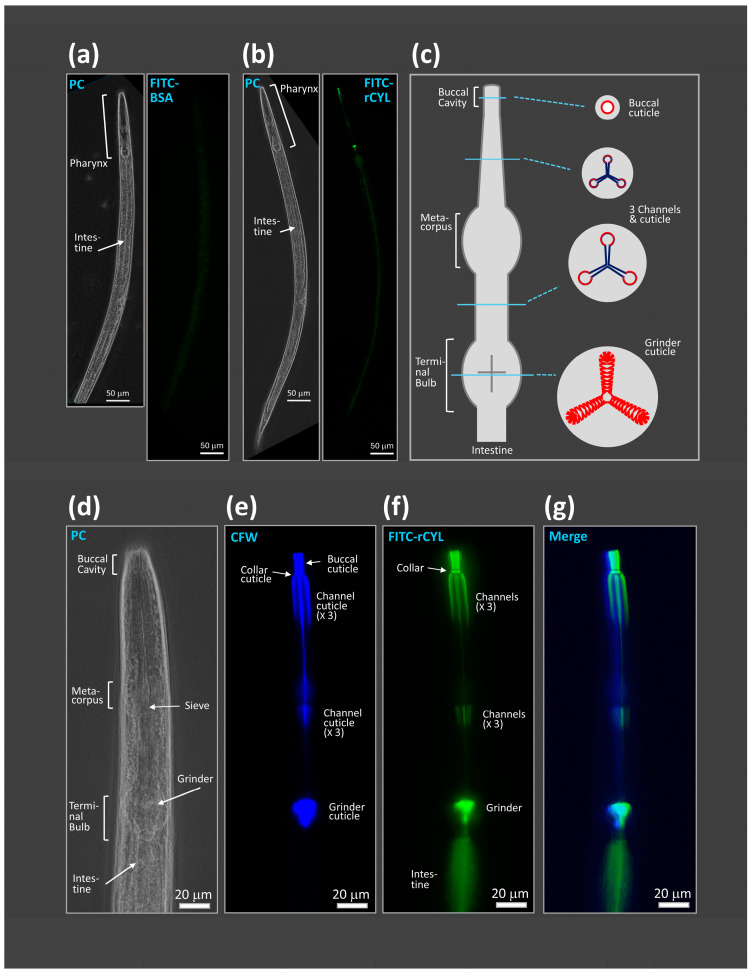
Localization of fluorescently labeled rCYL at the pharynx in nematodes. Adult nematodes were incubated on medium containing 0.2 mg/mL FITC-rCYL or 0.2 mg/mL FITC-BSA. (**a**) The nematode fed on FITC-BSA showed a faint fluorescent signal along the intestine (left, phase-contrast; right, fluorescence). Scale bar, 50 µm. (**b**) The nematode fed on FITC-rCYL exhibited intense fluorescence signals along the pharynx, in addition to a faint signal along the intestine. Scale bar, 50 µm. (**c**) Schematic presentation of the pharynx of *C. elegans*. The cuticle that lines the pharynx is indicated in red. (**d**) Phase-contrast image of the anterior part of the nematode that was fed on FITC-rCYL and then stained with Calcofluor white (CFW). Scale bar, 20 µm. (**e**) Fluorescence image detecting CFW in the same part of the nematode as in (**d**). CFW is a chitin probe. Scale bar, 20 µm. (**f**) Fluorescence image detecting FITC-rCYL in the same part of the nematode as in (**d**). Scale bar, 20 µm. (**g**) Images (**e**) and (**f**) were merged. FITC-rCYL localized at the structure that was stained with CFW. Scale bar, 20 µm.

**Figure 4 toxins-17-00118-f004:**
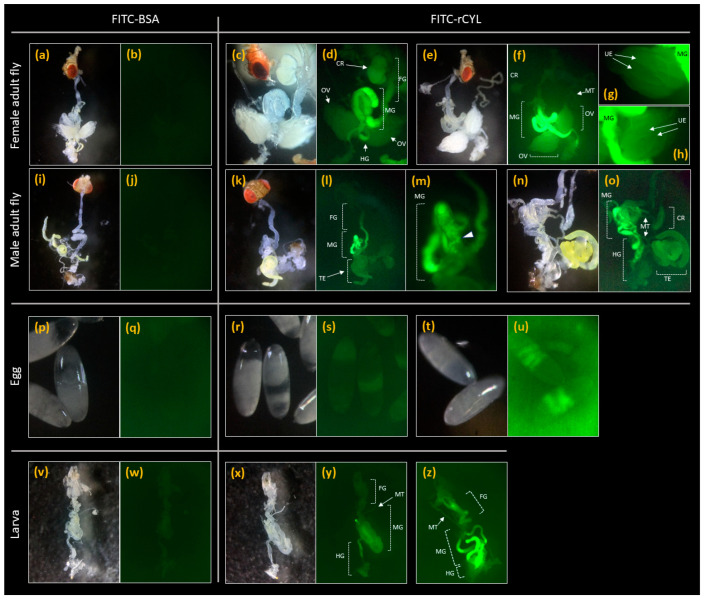
Localization of fluorescently labeled rCYL in *Drosophila*. Stereo-microscopic (**a**) and fluorescence (**b**) images of a female adult fly that was fed on medium containing FITC-BSA and dissected are shown, as a control. In (**c**–**h**), stereo-microscopic images (**c**,**e**) and fluorescence images (**d**,**f**–**h**) of female adult flies that were fed on medium containing FITC-rCYL and dissected, are shown. In (**d**,**f**), an intense fluorescent signal at the midgut (MG) and weak signals at the foregut (FG), hindgut (HG), crop (CR), ovary (OV), and malpighian tubules (MT) were observed. In (**g**,**h**), the ovary (OV) in (**f**) was expanded. Unfertilized eggs (UE) were also detected by a fluorescent signal. In (**i**,**j**), a stereo-microscopic image (**i**) and fluorescence image (**j**) of a male adult fly that was fed on medium containing FITC-BSA and dissected are shown as a control. In (**k**–**o**), stereo-microscopic images (**k**,**n**) and fluorescence images (**l**,**m**,**o**) of male adult flies that were fed on medium containing FITC-rCYL are shown. In (**l**,**m**,**o**), an intense fluorescent signal at the midgut (MG) and weak fluorescent signals at the foregut (FG), hindgut (HG), crop (CR), testis (TE), and malpighian tubules (MT) were observed. The image of the midgut in (**l**) is expanded in (**m**), revealing fluorescent particles (arrowheads) in the midgut. In (**p**,**q**), a stereo-microscopic image (**p**) and fluorescence image (**q**) of eggs laid by adult flies fed on medium containing FITC-BSA are shown as a control. In (**r**–**u**), stereo-microscopic images (**r**,**t**) and fluorescence images (**s**,**u**) of eggs laid by adult flies fed on medium containing FITC-rCYL are shown. Weak, unevenly distributed fluorescence signals (**s**) or, in some cases, multiple fluorescence bands around the center of the eggs (**u**) were observed. In (**v**,**w**), a stereo-microscopic image (**v**) and fluorescence image (**w**) of larvae that hatched from eggs laid by flies fed on medium containing FITC-BSA are shown as a control. In (**x**–**z**), a stereo-microscopic image (**x**) and fluorescence images (**y**,**z**) of larvae that hatched from the eggs of adult flies fed on medium containing FITC-rCYL and then grew on rCYL–medium are shown. In (**y**,**z**), an intense fluorescent signal at the midgut (MG) and weak fluorescent signals at the foregut (FG), hindgut (HG), and malpighian tubules (MT) were observed.

**Figure 5 toxins-17-00118-f005:**
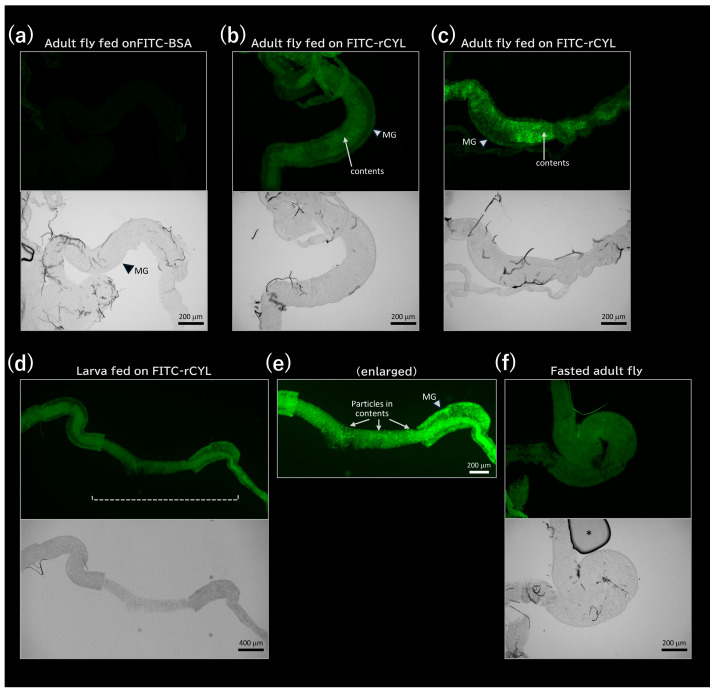
Fluorescence microscopic observation of the midgut contents. The adult flies fed on FITC-BSA (**a**) or FITC-rCYL (**b**,**c**) were dissected, fixed with paraformaldehyde, and then observed. In contrast to (**a**), fluorescent contents (arrows) were observed in the midgut (MG in **b**,**c**). The larvae fed on FITC-rCYL were also dissected, fixed, and then observed (**d**). The region with a broken line in (**d**) was expanded and brightened in (**e**) to clearly reveal the midgut (MG) tissue and the fluorescent contents. In (**e**), due to the partial breakdown of the midgut tissue, the contents were observed to be composed of many fluorescent particles (arrows). (**f**) The midgut of the adult fly that was fed on FITC-rCYL and then fasted in PBS showed no fluorescent contents. Asterisk: air bubble. Scale bars in (**a**–**c**,**e**,**f**): 200 µm. Scale bar in (**d**): 400 µm.

**Figure 7 toxins-17-00118-f007:**
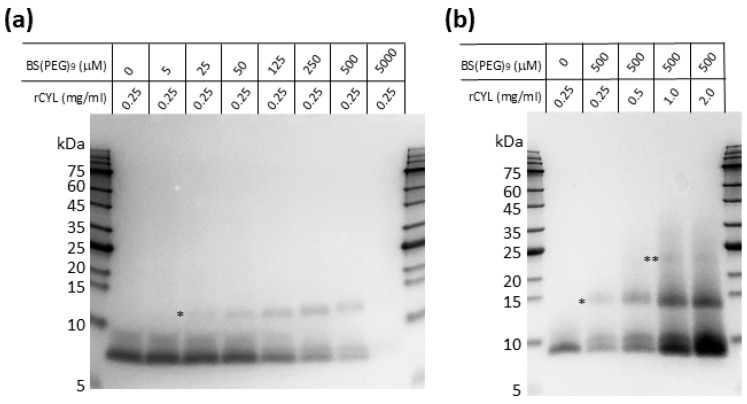
Protein cross-linking of rCYL. (**a**) rCYL was cross-linked via incubation with bis-*N*-succinimidyl-(nonaethylene glycol) ester [BS(PEG)_9_], and the reaction mixture was resolved by SDS-PAGE. The protein bands were detected using Coomassie brilliant blue staining. By increasing the concentration of BS(PEG)_9_ from 5 µM to 500 µM, a band representing the rCYL dimer was observed at ~13 kDa (indicated with an asterisk). When the cross-linker was employed at 5 mM, the protein bands disappeared from the gel. (**b**) As the concentration of rCYL increased in the presence of BS(PEG)_9_, two protein bands were observed at ~13 kDa (indicated with an asterisk) and ~24 kDa (indicated with the double asterisks), which represent the rCYL dimer and the rCYL tetramer, respectively.

**Figure 8 toxins-17-00118-f008:**
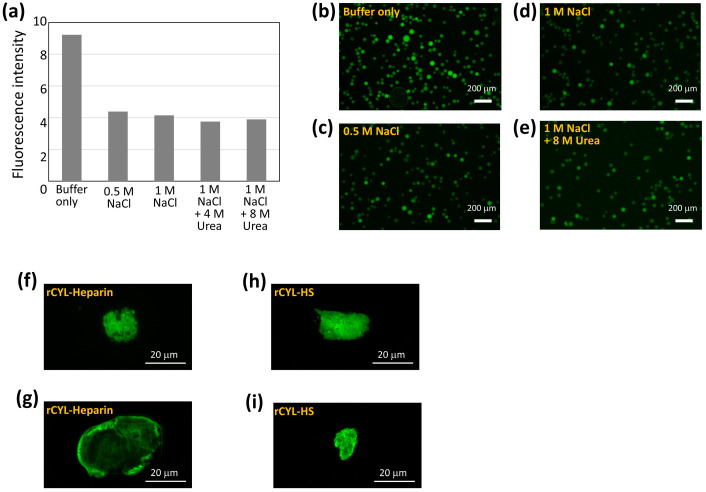
rCYL binding assay to heparin and heparan sulfate. (**a**) FITC-rCYL at 0.25 mg/mL was incubated with 100 µL of porcine–heparin agarose resin in 20 mM sodium phosphate buffer for 1 h. After washing the resin, the resin was incubated in 20 mM sodium phosphate (indicated as “Buffer only”), 20 mM sodium phosphate containing 0.5 M NaCl (“0.5 M NaCl”), 20 mM sodium phosphate containing 1 M NaCl (“1 M NaCl”), 20 mM sodium phosphate containing 1 M NaCl plus 4 M urea (“1 M NaCl + 4 M urea”), or 20 mM sodium phosphate containing 1 M NaCl plus 8 M urea (“1 M NaCl + 8 M urea”) for 10 min. After washing the resin, the fluorescence intensity of the resin was measured. (**b**–**e**) Representative fluorescence images of the heparin resins that were recovered after incubation in 20 mM sodium phosphate (indicated as “Buffer only”) (**b**), in the same buffer containing 0.5 M NaCl (“0.5 M NaCl”) (**c**), in the same buffer containing 1 M NaCl (“1 M NaCl”) (**d**), or in the same buffer containing 1 M NaCl plus 8 M urea (“1 M NaCl + 8 M urea”) (**e**). Scale bars, 200 µm. (**f**–**i**) rCYL at 0.4 mg/mL was incubated with 4 mg/mL heparin (**f**,**g**) or heparan sulfate (HS) (**h**,**i**) in 20 mM phosphate buffer containing 150 mM NaCl for 30 min, and was then incubated with 5 µM thioflavin T. Representative images of amyloid aggregates that emit green fluorescence are shown. Scale bars, 20 µm.

## Data Availability

The original contributions presented in this study are included in the article/Appendix A. Further inquiries can be directed to the corresponding author.

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
