# Peer review of "Cylindracin, a Fruiting Body-Specific Protein of Cyclocybe cylindracea, Represses the Egg-Laying and Development of Caenorhabditis elegans and Drosophila melanogaster"

_toxins, 2025, doi:10.3390/toxins17030118_

Round 1
Reviewer 1 Report
Comments and Suggestions for Authors
Reviewer comments for Toxins manuscript-345474
The manuscript, Cylindracin, a Fruiting Body-Specific Protein of Cyclocybe cylindracea, Represses the Egg-Laying and Development of Caenorhabditis elegans and Drosophila melanogaster, reports the effects of a small, cysteine-rich protein on development and reproduction of a model nematode and insect. Overall, the manuscript is well written and clear, although some areas for improvement are provided. Although rCYL does not appear to be very potent in its acute and chronic effects, its impact on egg laying and transgenerational effects are interesting.
A few areas for improvement will be described here, but comments are added within the pdf copy of the manuscript with more details and additional suggestions. First, some statistical analysis is needed for the data reported in section 2.1 and 2.2 for the effects of rCYL development and egg laying. Additionally, the biotoxicity testing scheme for evaluating rCYL on fly egg laying and transgenerational effects was confusing. I suggest that the authors consider revising the results section to make the testing more clear. As part of this, I recommend revising the schematic (“Scheme 1”) to include the duration of each step and additional steps showing how G2-2 and G3 were created. Scheme 1 should also be added to Figure 2.
The authors also report testing of the interaction of rCYL with heparin and heparin sulfate and conclude that rCYL binds to heparin “strong” and to heparin sulfate “with a lower affinity” forming a “high molecular weight complex” that is not denatured to allow migrate into gels by SDS-PAGE (lines 450-456). Although the results reported are consistent with these conclusions, the authors did not show any evidence for high MW complexes forming as the conc. of heparin was increased. Such complexes would be apparent in the wells of the gel which are not visible in the images of the gels that were provided (in Fig. S2b). Please provide gel images showing evidence for high MW complexes.
The authors proceed to evaluate the interaction of FITC-rCYL with heparin coupled agarose in the presence of high salt concentrations in the absence and presence of Urea and conclude that the interaction of rCYL with heparin is specific and partially involves ionic interactions (Fig. 8 a-e). However, the experiment as performed doesn’t reveal anything about specific recognition of heparin by rCYL or the affinity of the interaction. I suggest that the authors test the ability of unlabeled rCYL to prevent the interaction between FITC-rCYL and heparin agarose to show whether the interaction is specific and to determine an apparent affinity of the interaction under different salt conditions (starting at 0.05 M NaCl). Without measures of affinity, the authors should avoid referring to it in the Results and Discussion (i.e., avoid mentioning high, strong, tight, etc.) Next the authors incubate an unreported concentration of rCYL with heparin or heparin sulfate and then stain with thioflavin T to reveal aggregates drawing conclusions about fibril formation. However, even non-specific interactions with heparin/HS could cause similar results. The conclusion would be better supported by direct evidence of specific interaction between rCYL and heparin/HS such as the testing suggested above using FITC-rCYL and unlabeled rCYL.
Finally, in the discussion, the authors refer to data not shown twice. This should be avoided since the reader has no information about the experimental details or the data to consider the significance of the statements.
Please refer to pdf copy for additional line-by-line comments/suggestions.

Reviewer 2 Report
Comments and Suggestions for Authors
This manuscript offers detailed experiments on the mode of action of rCYL protein, a recominant protein associated with the cysteine-rich domain of the fungal protein cylindracin. Experiments clearly demonstrate biological activity against C. elegans and D. melanogaster. An array of experiments point to interference with normal digestive properties in both organisms, albeit with different specific modes of action. This mode of action (decreased digestive ability leading to malnutrition) seems to agree with experimental results involving organismal weight and survivorship. I found no problems in any of the experiments nor did I see any flaws in the authors' logic in explaining their results. The wide array of experiments were thorough and convincing in my view. I have only two points for minor revision. First, I notice that scientific names are not properly italicized in some of the references. Second, in the discussion I think it is important for the authors to consider the apparent lack of specificity of the effect of rCYL, given that similar modes of action (malnutrition) occurred in very diverse taxa. The issue is that as a potential control method, would or could rCYL interfere with beneficial insects. My suspicion is that it would not, as it would be necessary for rCYL to accumulate in biological meaningful doses when pest insects were eaten by predators and parasite. Nevertheless, I think this issue merits mention in the discussion.
Reviewer 3 Report
Comments and Suggestions for Authors
The overall language and presentation of the work titled "Cylindracin, a fruiting body-specific protein of Cyclocybe cylindracea, represses the egg-laying and development of Caenorhabditis elegans and Drosophila melanogaster" were exceptionally engaging. However, the conclusions included references to certain results derived from unpublished data. To enhance the manuscript's integrity and focus, it would be advantageous to rephrase this section, eliminating any references to unpublished research.
Round 2
Reviewer 1 Report
Comments and Suggestions for Authors
2nd Round reviewer comments for Toxins manuscript-345474
Manuscript: Cylindracin, a Fruiting Body-Specific Protein of Cyclocybe cylindracea, Represses the Egg-Laying and Development of Caenorhabditis elegans and Drosophila melanogaster
The authors have adequately addressed most of my queries and comments. One exception is their response to my comments related to their experimental results reported in Fig. 8 (Comment 5; and comments related to lines 471-474 and lines 479-482) where FITC-rCYL is tested for interactions with heparin and heparin sulfate resins. Regardless, the authors have now provided the concentrations of FITC-rCYL used in those experiments (~40 and 60 micromolar; very high for fluorescently-labeled proteins), and therefore, the astute reader can decide whether anything meaningful can be concluded about the interactions whether specific or not. Slight editing of new text added to the discussion suggested below.
Revised MS lines 605-610:
Suggest to edit or replace with: rCYL might interfere with the growth of beneficial insects that are predatory to pest insects that have consumed rCYL. However, large quantities of intoxicated pest insects would likely need to be consumed to accumulate biologically meaningful doses within beneficial insects. To further clarify the impact of rCYL against various predators, assay systems that can be performed under near-natural conditions are required.
